# Exploring Exosome Contributions to Gouty Arthritis: A Proteomics and Experimental Study

**DOI:** 10.3390/ijms26115320

**Published:** 2025-06-01

**Authors:** Chengjin Lu, Xiaoxiong Yang, Xue Wang, Yu Wang, Bing Zhang, Zhijian Lin

**Affiliations:** 1Department of Clinical Chinese Pharmacy, School of Chinese Pharmacy, Beijing University of Chinese Medicine, Beijing 100029, China; 20230935117@bucm.edu.cn (C.L.); yangxiaoxiong1539@163.com (X.Y.); wangxue20220222@163.com (X.W.); wangyuxh@163.com (Y.W.); zhangb@bucm.edu.cn (B.Z.); 2Research Center for Pharmacovigilance and Rational Use of Chinese Medicine, Beijing University of Chinese Medicine, Beijing 100029, China

**Keywords:** proteomics technology, experimental validation, exosomes, gouty arthritis, pathological basis

## Abstract

This study investigated the role of exosomes in the pathological processes of gouty arthritis (GA), with the aim of clarifying their mechanistic role and pathological significance in the onset and progression of GA. Using a rat model of GA established through potassium oxonate and yeast gavage combined with intra-articular monosodium urate (MSU) injection, we isolated and characterized plasma exosomes using transmission electron microscopy (TEM), nanoparticle tracking analysis (NTA), and Western blotting. Differential exosomal protein expression was analyzed using 4D label-free proteomics technology, followed by GO and KEGG enrichment analyses, and protein–protein interaction (PPI) network construction to identify core targets. In vivo experiments measured the expression levels of CTSD in synovial tissues and joint fluid, as well as HPRT1 in renal tissues, while in vitro experiments involved co-culturing NRK cells with exosomes to validate target protein expression. The results indicated that serum uric acid levels were significantly elevated in the model group (*p* < 0.01), accompanied by pronounced joint swelling and inflammation. Exosome characterization confirmed their typical bilayer membrane structure and the expression of marker proteins (CD63/TSG101). Proteomic analysis identified 40 differentially expressed proteins (12 upregulated and 28 downregulated) enriched in pathways such as complement and coagulation cascades, autophagy, lysosomal function, and purine metabolism. In vivo and in vitro experiments demonstrated significantly increased CTSD expression (*p* < 0.05/*p* < 0.01) and decreased HPRT1 expression (*p* < 0.05/*p* < 0.01) in the model group, suggesting that exosomes are involved in the occurrence and development of GA by regulating purine metabolism and lysosomal dysfunction. These findings offer new insights into disease mechanisms and potential therapeutic targets.

## 1. Introduction

Gouty arthritis (GA) is an inflammatory joint disease caused by hyperuricemia, characterized by the deposition of monosodium urate (MSU) crystals within the joints, which triggers severe inflammatory reactions and pain [1]. The burden of GA in China is increasing as a significant subtype of metabolic rheumatic disease, with the latest epidemiological data showing that the number of patients has reached 14.66 million, with an overall prevalence of 1.1%. Approximately 40% of patients experience multiple acute attacks annually, making it the second most common metabolic disease after diabetes [2]. This continuously growing trend in incidence and its significant impact on patients’ quality of life urgently requires us to delve into its pathogenesis and develop new therapeutic strategies.

It is worth noting that while hyperuricemia is a necessary prerequisite for the development of GA, only 36% of patients with hyperuricemia progress to clinical gout [3]. This clinical heterogeneity suggests the existence of a complex regulatory mechanism. Recent imaging studies have detected MSU crystal deposition and subclinical joint damage in the joints of asymptomatic hyperuricemic patients, further indicating the existence of inter-organ signaling between uric acid metabolic disorders and joint lesions [4]. In this process, exosomes may play a significant role as key carriers of intercellular communication. These membrane vesicles, with diameters ranging from 30 to 150 nm, are enveloped by a lipid bilayer and carry biologically active molecules, such as proteins, nucleic acids (including miRNAs), and lipids, enabling long-distance information transmission through the body’s fluid circulation, thereby regulating the functional state of receptor cells [5].

Recent studies have shown that exosomes may participate in the pathogenesis of GA through multiple pathways. In gout models, neutrophils lacking vesicle fusion ability exhibit reduced migration activity [6], whereas MSU-stimulated neutrophil exosomes can inhibit osteoblast function by delivering miR-1246, accelerating bone erosion [7]. Another study confirmed that exosomes carrying inflammasomes can directly activate the NF-κB signaling pathway, triggering an inflammatory cascade reaction [8]. XiuMin et al. discovered characteristic changes in miRNA expression profiles in plasma exosomes from GA patients, with five miRNAs, including hsa-miR-151a-3p and hsa-miR-199a-5p, which are potentially involved in disease onset [9]. Synovial fluid analysis showed that the numbers of exosomes in the joint cavities of GA patients were significantly higher than those of osteoarthritis patients, but lower than those of rheumatoid arthritis patients; these exosomes still promote osteoclast differentiation without classical inducers, but their osteoclastic activity is weaker than that of rheumatoid arthritis patients [10]. Yukai Huang et al. identified 69 differentially expressed proteins in exosomes from the synovial fluid of GA patients through proteomics, among which 25 highly expressed proteins, such as lysozyme C and S100-A9, are associated with immune processes like neutrophil degranulation and have potential as biomarkers [11]. Additionally, iron death-related proteins (ACSL4, VDAC2, etc.) in urinary exosomes from acute-phase GA patients are significantly upregulated, possibly exacerbating inflammatory responses by promoting lipid peroxidation and reactive oxygen species accumulation [12]. Together, these findings reveal the multidimensional regulatory role of exosomes in GA pathogenesis and provide a new perspective for understanding disease mechanisms.

To further explore the multifaceted roles of exosomes in the pathogenesis of gouty arthritis (GA), this study employed a comprehensive approach that combined four-dimensional unlabeled proteomics with experimental validation. We established a rat model of gouty arthritis by administering potassium oxalate via gastric lavage, yeast administration, and intra-articular injection of sodium urate (MSU) crystals to simulate the pathological state of hyperuricemia and MSU crystal deposition. Using transmission electron microscopy (TEM), nanoparticle tracking analysis (NTA), and Western blotting, we isolated and identified plasma exosomes. Through proteomic analysis, we screened for differentially expressed proteins in exosomes and identified key targets while also conducting in vitro and in vivo functional validation experiments (Figure 1). This study aimed to elucidate the specific roles of exosomes in the pathogenesis of gouty arthritis and provide a theoretical basis for the development of new therapeutic targets.

## 2. Results

### 2.1. Establishment of Rat Model with Hyperuricemia and Gouty Arthritis

Serum uric acid levels serve as the basis for the diagnosis of hyperuricemia. Compared to the control group, the serum uric acid levels in the model group were significantly elevated (Figure 2A), indicating successful induction of the hyperuricemia rat model. Additionally, compared to the control group, ankle joint swelling in the model group of rats was significantly increased at all time points (Table 1, Figure 2B). The ankle joint tissue structure in the model group of rats was clearer than that in the control group, with a normal arrangement of synovial cells and no cell proliferation, inflammatory infiltration, or vascular congestion. In contrast, the ankle joint synovial tissue in the model group showed significant inflammatory infiltration and vascular congestion (Figure 2C), suggesting the successful use of Coderre to replicate an acute gouty arthritis attack.

### 2.2. Characterization of Plasma Exosomes in Rats

Exosomes were extracted from rat plasma using the EVtrap method and then characterized using transmission electron microscopy (TEM, HT-7700, Hitachi, Tokyo, Japan), nanoparticle tracking analysis (NTA, N30E, Xiamen Flow Bioscience Co., Ltd., Xiamen, China), and Western blotting (WB). TEM revealed that plasma exosomes had uniform sizes, presenting as lipid bilayer vesicles with a round or oval shape, consistent with typical exosome morphology (Figure 3A). NTA detected that the exosomes separated by EVtrap magnetic beads had an average particle size consistent with classical exosome measurements and the exosomes from the model group had a smaller average particle size than those from the control group (Table 2, Figure 3B). WB detected specific proteins CD63 and TSG101 in the exosomes, while the negative protein Calnexin was not detected. Conversely, the negative protein Calnexin did not appear in the detection, further confirming that these exosomes exhibit typical exosome characteristics and have high purity (Figure 3C).

### 2.3. Proteomics Analysis of Plasma Exosomes in Rats

To comprehensively investigate the proteomic differences between exosomes derived from the control and model groups, we employed four-dimensional (4D) label-free quantitative proteomics technology. The quality control results of mass spectrometry analysis are shown in Figure 4A, identifying a total of 8511 peptides, among which 7497 were unique. These peptides were mapped to 1369 protein groups, with 875 protein groups containing quantifiable data (Figure 4B). To evaluate the reproducibility of samples within groups, we conducted Venn diagram analysis, which revealed 615 overlapping proteins in the control group and 449 overlapping proteins in the model group (Figure 4C). Using thresholds of fold change (FC) > 2 (or <0.5) and a *p*-value < 0.05, we identified 40 significantly differentially expressed plasma exosomal proteins in the model group versus the controls (12 upregulated and 28 downregulated; Table 3, Figure 4D). In analyzing group-specific proteins, the screening criteria for proteins were as follows: in one group, half or more of the samples had valid values (non-null values), whereas in the other group, all data were null values. Based on this criterion, 16 upregulated differentially expressed proteins and 40 downregulated differentially expressed proteins were identified. These proteins were not displayed in the volcano plot or cluster analysis plot, but they will undergo subsequent bioinformatic functional analysis along with other proteins exhibiting significant differences. A hierarchical clustering algorithm was applied to classify the differentially expressed proteins between the two groups, and the clustering results were visually presented via a heatmap. Based on the clustering analysis, we observed high similarity in data patterns within groups, but low similarity between groups, which effectively distinguished the different groups (Figure 4E).

### 2.4. Analysis of Protein Domains in Plasma Exosomes of Rats

Protein domains are regions within larger protein molecules where adjacent supersecondary structures on the polypeptide chain are closely linked, forming two or more distinct local areas that can be clearly distinguished in space. Each domain typically consists of dozens to hundreds of amino acid residues, each with its own unique spatial structure, and is responsible for different biological functions. Domain prediction is crucial for studying key functional regions of proteins and their potential biological roles. Analysis of differential protein domains (Figure 5) showed that the structural domains of differential proteins between the model and control groups were mainly concentrated in immunoglobulin V-set domains, protein kinase domains, trypsin, immunoglobulin C1-set domains, cytoskeletal regulatory complex EF hand proteins, and ATP enzyme families (AAA).

### 2.5. GO Annotation and KEGG Enrichment Analysis of Exosome Proteins in Rat Plasma

To better understand the functional roles of plasma exosome proteins in both the control and model rat groups, we conducted comprehensive GO annotations and KEGG pathway enrichment analyses (Figure 6A–C). The differentially expressed proteins between the model and control groups were primarily enriched in biological processes, including branched-chain amino acid transport, iron ion balance, and carboxylic acid transmembrane transport. In terms of cellular components, the main enriched entries were endosomal vesicles, early endosomes, lysosomal membranes, and endocytosis. The primary enriched entries for molecular functions were guanosine nucleotide binding, purine nucleoside binding, and purine riboside binding. After comparing the differentially expressed proteins with the KEGG database for differential signaling analysis, it was found that, compared to the control group, the main pathways enriched by upregulated differential proteins in the plasma exosomes of the model group were the complement and coagulation cascade, autophagy, lysosomes, and apoptosis. The main pathways enriched by downregulated differential proteins were proteasome degradation, iron metabolism disorders, and the IL-10 anti-inflammatory signaling pathway. Other related enriched pathways also included the purine metabolic pathway (Figure 6D–E).

### 2.6. Protein Interaction Network Analysis of Exosome Proteins in Rat Plasma

Protein–protein interaction networks were constructed for all differentially expressed plasma exosomal proteins between control and model rats using STRING database version 12.0 and Cytoscape 3.7.2 (Figure 7), and the constructed PPI network consisted of 52 nodes and 188 edges (Figure 7A). Centrality metrics were calculated using the CytoHubba impl-0.1plugin to identify the top 20 core targets (Figure 7B), including the transferrin receptor (Tfrc), hypoxanthine-guanine phosphoribosyltransferase (Hprt1), iron reductase Steap3, GTPase Arf6, cathepsin D (Ctsd), transforming growth factor-β1 (Tgfb1), transcription factor Stat3, and kallidin-1 (Kng1). These functions involve metabolism and transport, iron and heme homeostasis, vesicle transport, membrane fusion, lysosomes, and protein degradation. Based on the results of GO and KEGG enrichment analyses, CTSD, which is associated with apoptosis and autophagy, and HPRT1, which is related to purine metabolism, were selected as key targets for subsequent research.

### 2.7. In Vitro and In Vivo Experiments Were Conducted to Verify the Results

#### 2.7.1. In Vivo Experiments

Using Elisa technology, the expression levels of HPRT1 and CTSD proteins in kidney tissue, synovial tissue, and joint fluid were verified in both the control and model groups. During the experiment, compared to the control group, the expression levels of CTSD in the synovial tissues and joint fluid of the model rats were significantly increased (*p* < 0.01), whereas the expression levels of HPRT1 in kidney tissue were significantly decreased (*p* < 0.01). This suggests that the HPRT1 and CTSD proteins are key targets involved in the development of gouty arthritis, see (Table 4 and Figure 8).

#### 2.7.2. In Vitro Experiments

##### Cell Morphology

During their adhesion and growth process, NRK cells could be observed on the 1st day after passage to have most cells adhering to the wall, with only a few cells adhering in multiple forms. From the 2nd to 3rd days, cell growth was well maintained; they were uniformly sized, firmly attached to the wall, and had clear boundaries, maintaining a relatively intact morphology that exhibited epithelial characteristics. Additionally, the cell nuclei were clearly visible and arranged neatly and closely together. See Figure 9.

##### Effects of Plasma Exosomes on the Content of CTSD and HPRT1 Proteins in NRK Cells

Using WB, the changes in CTSD and HPRT1 protein expression in NRK cells after plasma exosome treatment in both the control and model groups was verified. Compared to the results of plasma exosome treatment in the control group, plasma exosome treatment in the model group significantly increased the expression of CTSD protein in NRK cells (*p* < 0.05) and significantly decreased the expression of HPRT1 protein (*p* < 0.05), as shown in the Table 5 and Figure 10. This further suggests that exosomes, via intercellular communication, influence the expression of HPRT1 and CTSD proteins in recipient cells, participating in the development and progression of gouty arthritis.

## 3. Discussion

This study successfully constructed a rat model of gouty arthritis (GA) using potassium oxazoline acid combined with yeast gastric administration and the Coderre method. The model not only exhibited significant elevation in serum uric acid levels and ankle joint swelling but also confirmed typical pathological features such as synovial inflammation infiltration and vascular congestion through pathological analysis (Figure 2). This model simulates the core pathological characteristics of human GA [13], in which purine metabolic disorders lead to urate crystal deposition, providing a reliable platform for subsequent mechanistic studies.

To address the technical challenges of exosome isolation, this study employed EVtrap magnetic bead technology, which achieves high purity and efficiency through the specific binding of hydrophilic/lipophilic groups to the exosome membrane [14,15]. Given the limited sample size of rat joint fluid, which was insufficient for omics analysis, we opted for plasma as the source of exosomes because it is rich in disease-related protein markers and can systematically reflect pathological conditions [16]. According to the MISEV2018 guidelines [17,18], we conducted multimodal validation of the isolated exosomes. Transmission electron microscopy revealed typical biconvex vesicle structures with average particle sizes of 160.1 nm and 118.6 nm, respectively. Western blotting detected the positive markers TSG101 and CD63, while Calnexin was negative (Figure 3), confirming the high purity and integrity of the exosomes, thus meeting the requirements for downstream analysis.

Using 4D unlabeled quantitative proteomics technology, we systematically analyzed proteomic differences between plasma exosomes from control and GA model rats (Figure 4). A total of 1369 proteins were identified, of which 875 were quantifiable. Differential expression analysis revealed 40 significantly differentially expressed proteins (12 upregulated and 28 downregulated) and 56 group-specific proteins (16 upregulated and 40 downregulated). Hierarchical clustering analysis showed high intragroup consistency and significant intergroup separation, supporting the reliability of the research results. Domain analysis revealed key functional modules (Figure 5), including immunoglobulin V-set, protein kinase, trypsin-like, and AAA ATP enzyme domains, suggesting abnormalities in immune regulation, proteolysis, and energy metabolism. Functional enrichment analysis further indicated that the differentially expressed proteins mainly participated in branched-chain amino acid transport, iron homeostasis, vesicle transport, guanosine nucleotide binding, purine nucleotide binding, and purine riboside binding. KEGG pathway analysis showed that upregulated proteins were associated with complement and coagulation cascades, autophagy, lysosomes, and apoptosis, whereas downregulated proteins were related to proteasome degradation, iron metabolism disorders, and IL-10 anti-inflammatory signaling (Figure 6). Consistent with previous studies, monosodium urate (MSU) induces lysosomal damage and activates the macrophage NLRP3 inflammasome, thereby mediating the production of inflammatory factors and affecting the ratio of Tregs/Th17 cells, contributing to the pathogenesis of GA [19]. Additionally, some studies have found that serum ferritin levels are positively correlated with gout attacks, and serum ferritin levels in patients with gout were significantly higher than those in the control group [20]. Second, purine metabolism was identified as a core pathway, which is highly consistent with the pathogenesis of gout driven by hyperuricemia.

Protein interaction network (PPI) analysis revealed core hub proteins such as Tfrc, HPRT1, Steap3, Arf6, Ctsd, Tgfb1, Stat3, and Kng1, indicating that iron metabolism, lysosomal function, and inflammatory signaling play crucial roles in the progression of GA (Figure 7). Among these, the abnormal expression of CTSD and HPRT1 is particularly noteworthy. As a rate-limiting enzyme in the purine salvage pathway, the downregulation of HPRT1 can impair this pathway, leading to purine metabolic disorders and increased uric acid production. Defects in HPRT1 have been widely associated with hereditary gout (such as Lesch-Nyhan syndrome) [21,22,23,24,25]. In this study, we found that the expression level of HPRT1 in the kidney tissue of gout model rats was significantly lower than that of the control group (Figure 8). When exosomes from gouty arthritis rats were co-cultured with NRK cells, the protein expression of HPRT1 in NRK cells showed a significant downward trend compared to that in the normal rat plasma exosome treatment group (Figure 10). These results suggest that the downregulation of HPRT1 expression in local tissues may reflect an acquired metabolic disorder, providing a new perspective on the pathogenesis of gout and suggesting that HPRT1 dysfunction in patients with gout may spread through intercellular communication mediated by exosomes rather than systemic enzyme deficiency. In contrast, CTSD is a lysosomal protease and its expression was significantly elevated in the synovial tissue and joint fluid of GA model rats (Figure 7). Exosome co-culture experiments have confirmed that exosomes derived from the plasma of rats with gouty arthritis can induce upregulation of CTSD in NRK cells, indicating the critical role of exosomes in transmitting signals of lysosomal dysfunction (Figure 9). The upregulation of CTSD may reflect the excessive activation of lysosomes in GA, leading to tissue destruction and the release of inflammatory factors. Previous studies also support this view, demonstrating significantly elevated expression of tissue protease D in synovial fluid from patients with gouty arthritis [26] and significant changes in the expression profile of lysosomal-related proteins (such as CTSZ, AP1B1, and LAMP2) in urinary exosomes from acute gout attacks [27]. At the molecular level, CTSD is released from lysosomes into the cytoplasm, cleaving Bid to generate tBid, which then activates Bax, prompting the mitochondrial release of cytochrome C, ultimately triggering the caspase-9/3 apoptosis pathway [28]. MSU crystals can cause lysosomal damage through exosome-dependent or non-exosome-dependent pathways, inducing autophagy impairment mediated by CTSD, promoting macrophage apoptosis, and exacerbating inflammatory responses [29]. These studies collectively highlight the significant role of exosome-mediated purine metabolic disorders and lysosomal dysfunction in the progression of gout, laying the theoretical foundation for CTSD and HPRT1 as biomarkers for diagnosing gouty arthritis clinically and developing therapeutic strategies targeting exosomes.

## 4. Materials and Methods

### 4.1. Materials

#### 4.1.1. Animals

Twenty-four male Sprague Dawley (SD) rats (SPF grade, specific pathogen-free), weighing 150 ± 10 g, were provided by Sibefu Biotechnology Co., Ltd. (Beijing, China, License No.: SCXK (Jing) 2019-0010). The animal experiments were approved by the Bioethics Committee (Approval No.: BUCM-4-2022092601-3113).

#### 4.1.2. Cells

Normal rat kidney cells (NRK cells) were purchased from the Cell Bank of the Committee on Type Culture Collection of the Chinese Academy of Sciences.

#### 4.1.3. Main Reagents

Potassium oxonate (Yuanye Bio-Technology Co., Ltd., Shanghai, China), sodium urate (Sigma, St. Louis, MO, USA), yeast extract powder (Thermo Fisher Scientific, Waltham, MA, USA), uric acid assay kit (Zhongsheng Beikong Bio-Technology Co., Ltd., Beijing, China), sodium carboxymethyl cellulose (CMC-Na) (Shanghai Yuanye Bio-Technology Co., Ltd., Shanghai, China), a hematoxylin and eosin (H&E) staining kit (Wuhan Servicebio Technology Co., Ltd., Wuhan, China), CD63 antibody (Santa Cruz Biotechnology, Dallas, TX, USA), TSG101 antibody (Proteintech, Chicago, IL, USA), Calnexin mouse monoclonal antibody (Proteintech, Chicago, IL, USA), Goat anti-mouse IgG (H + L) (Proteintech, Chicago, IL, USA), Goat anti-rabbit IgG (H + L) (Proteintech, Chicago, IL, USA), Hprt1 polyclonal antibody (Proteintech, Chicago, IL, USA), Cathepsin D polyclonal antibody (Proteintech, Chicago, IL, USA), DMEM basal medium (Gibco, NY, USA), penicillin–streptomycin solution (double antibiotics) (Beijing Aoking Bio-Technology Co., Ltd., Beijing, China), and premium fetal bovine serum (FBS) (Beijing Aoking Bio-Technology Co., Ltd., Beijing, China) CTSD Elisa Kit (Jiangsu Enzyme Immunity Industrial Co., Ltd., Yancheng, China), HPRT1 Elisa Kit (Jiangsu Enzyme Immunity Industrial Co., Ltd., Yancheng, China), BCA Protein Concentration Detection Kit (Beijing Solarbio Science & Technology Co., Ltd., Beijing, China).

#### 4.1.4. Major Instruments

Microfuge 20R benchtop refrigerated centrifuge (Beckman Coulter, CA, USA), CP100MX ultracentrifuge (Hitachi, Tokyo, Japan), HT-7700 transmission electron microscope (TEM) (Hitachi, Tokyo, Japan), N30Enanoparticle tracking analyzer (NTA) (Xiamen Flow Bioscience Co., Ltd., Xiamen, China), SHZ88-1 thermostatic water bath (Taicang Guangming Experimental Instrument Factory, Taicang, China), 3K1S refrigerated high-speed centrifuge (Sigma, Osterode, Germany), Sunrise microplate reader (TECAN, Männedorf, Switzerland), DT5-3 low-speed benchtop centrifuge (Beijing Centrifuge Co., Ltd., Beijing, China), WD-9405C decolorizing shaker (Beijing Liuyi Biotechnology Co., Ltd., Beijing, China), DYY-6D electrophoresis system (Beijing Liuyi Biotechnology Co., Ltd., Beijing, China), Trans-Blot Turbo rapid semi-dry transfer system (Bio-Rad, Hercules, CA, USA), MiniChemi 500 automated chemiluminescence imaging system (Beijing Saizhi Innovation Technology Co., Ltd., Beijing, China), a super-resolution tissue imaging system (Leica, Wetzlar, Germany), C170 CO_2_ incubator (Binder, Tuttlingen, Germany), Olympus BX43 biological microscope (Olympus, Tokyo, Japan), LDZM vertical autoclave sterilizer (Shanghai Shen’an Medical Equipment Factory, Shanghai, China), Vortex 2 vortex mixer (IKA, Staufen, Germany), and DT5-3 low-speed benchtop centrifuge (Beijing Centrifuge Co., Ltd., Beijing China).

### 4.2. Methods

#### 4.2.1. Animal Grouping and Model Preparation

After an adaptive feeding period, the SD rats were randomly divided into a control group and a model group based on body weight, with 12 rats in each group. During the experimental period, the model was established using a combination of potassium oxonate and yeast following the classic Coderre method [30]. From day 1 to day 10 of the experiment, the rats were administered 750 mg/kg potassium oxonate solution and 1 g/kg yeast solution via intragastric gavage. On day 8 of the experiment, a sterile syringe was used to inject a 25 mg/mL monosodium urate (MSU) suspension into the ankle joint cavity of the model group rats. The control group received an equal volume of physiological saline injected into the ankle joint cavity in the same manner. Successful injection was confirmed by the appearance of swelling on the contralateral side of the ankle joint.

#### 4.2.2. Sample Collection and Processing

On day 7 of the experiment, after 12 h of fasting (with free access to water), blood was collected from the tail vein. Blood samples were allowed to stand for 2 h and then centrifuged (3500 rpm, 10 min) to separate the serum. Joint circumference was measured before the ankle joint injection and at 4, 6, 12, and 24 h after the injection. At 24 h post-injection, the rats were anesthetized, and blood was collected from the abdominal aorta to isolate the plasma. The ankle joint tissue was fixed in 4% paraformaldehyde and the kidney tissue was frozen at −80 °C. The joint cavity was opened and rinsed with physiological saline to obtain lavage fluid, and the synovial tissue was stored at −80 °C.

#### 4.2.3. Detection Indicators and Methods

##### Indicators of Successful Model Induction

Serum uric acid levels were measured using the uricase method. A 2 cm strip was used to measure the circumference of the affected ankle joint in each group of rats, which was repeated three times. The joint swelling degree at each time point was calculated according to the formula. Hematoxylin–eosin (HE) staining was performed to observe pathological changes in the ankle joint.(1)$$\mathrm{Joint}\;\mathrm{swelling}\;\mathrm{rate}\;(\%)=\frac{Joint\;circumference\;at\;a\;given\;time\;point\;-\;Baseline\;joint\;circumference}{Baseline\;joint\;circumference}\times 100\%$$

##### Isolation and Identification of Plasma Exosomes

Plasma Exosome Separation (EVtrap Method)

Using a 200 μL plasma sample as an example, the sample was thawed at 37 °C. Next, 200 μL of plasma was added to 1 mL loading buffer solution, and 25 μL of magnetic EVtrap beads was added, inverted, mixed well, and incubated at room temperature for 1 h. After incubation, the solution was magnetically separated and the supernatant was collected and retained. Add washing buffer solution to the EP tube, aspirate to suspend the magnetic beads, and transfer the washing buffer solution containing the magnetic beads to the designated plate wells according to Table 6. Extraction was performed automatically following the default procedure.

Plasma Exosome Identification

(1) TEM analysis: 10 μL of the exosome sample is deposited on a copper grid and allowed to settle for 1 min. Absorb excess with filter paper, add 10 μL of uranyl acetate, and settle for 1 min. Dry at room temperature and image with 100 kV electron microscope. (2) Particle size analysis: Dilute 30 μL exosomes to 120 μL, mix 30 μL with 20 μL fluorescent antibodies (CD63, CD81), and incubate in the dark at 37 °C for 30 min. Add 1 mL cold PBS, centrifuge at 4 °C, 110,000× *g* for 70 min, discard supernatant, repeat once, and resuspend in 50 μL cold 1× PBS for measurement. (3) Western blot analysis: Mix proteinase inhibitor with RIPA buffer (1:100), and the exosomes are added and incubated on ice for 10 min. Centrifuge at 10,000× *g*, 4 °C for 5 min, transfer supernatant, mix with 5× loading buffer, heat at 100 °C for 10 min, and detect CD63, TSG101, and Calnexin.

Proteomics Analysis

(1) Exosome protein extraction and concentration determination: The exosome solution was mixed with SDT lysis buffer at 37 °C and incubated on ice for 30 min while stirring. A BCA standard solution was prepared and the sample was evenly mixed with the BCA mixture and incubated at 37 °C for 30 min. Absorbance at OD562 nm was measured using a microplate reader, and the protein concentration was calculated using a standard curve. (2) SDS-PAGE Separation: Take 20 μg of protein, add 5× loading buffer, denature in boiling water for 5 min, perform electrophoresis at 180 V for 45 min, and visualize the separation using Coomassie Brilliant Blue R-250 staining. (3) Peptide digestion: An appropriate amount of protein sample was reduced with 100 mM DTT, ultrafiltration centrifuged with UA buffer (8M urea + 150 mM Tris-HCl, pH 8.5), incubated with IAA in the dark at room temperature, and then treated with UA buffer and NH_4_HCO_3_ solution. Trypsin buffer (4 μg of trypsin dissolved in 50 mM NH_4_HCO_3_) was added for overnight digestion. The next day, the supernatant was collected, desalized using a C18 column, and dissolved in 0.1% formic acid for quantification at 280 nm. (4) LC-MS/MS Data Acquisition: The NanoElute system separates the hydrolyzed samples at nanoflow rates. The liquid chromatography phase uses 0.1% formic acid in water (Phase A) and 0.1% formic acid in acetonitrile/water (Phase B), with separation on a C18 reversed-phase analytical column. For mass spectrometry analysis, the timsTOF Pro mass spectrometer was operated in positive ion mode with a source voltage of 1.5 kV, detecting MS and MS/MS in a scan range of 100–1700 *m*/*z* using PASEF mode with an ion mobility range of 0.6–1.6 Vs/cm^2^. After primary MS acquisition, 10 secondary MS spectra were collected in PASEF mode, with a dynamic exclusion time of 24 s. (5) Mass Spectrometry Quality Control: The timsTOF Pro mass spectrometer precisely controls the mass deviation to generate high-quality MS1 and MS2 spectra. The identified peptide mass deviations were concentrated within 10 ppm. The Andromeda tool scores MS2 spectra. The 4D label-free data screening criteria were Peptide FDR ≤ 0.01 and Protein FDR ≤ 0.01. (6) Database Search: MaxQuant software(1.6.14) searches for secondary MS data against the Ensembl_Rattus_45936_20220121 database, incorporating a reverse database for FDR estimation and adding common contaminant databases. Trypsin was selected as the digestion enzyme with a maximum of two missed cleavages. The primary ion mass tolerance was set to 6 ppm (initial search: 20 ppm) and the secondary fragment ion mass tolerance was 20 ppm. Modifications: fixed carbamidomethylation at the C-terminus and variable oxidation at the M-terminus. The protein and peptide FDR thresholds were ≤0.01. (7) Bioinformatics Analysis: MaxQuant 1.6.14 analyzes MS data. The ComplexHeatmap R package (R 3.4) performs clustering analysis, while InterProScan 5.25-64.0 conducts domain analysis. GO and KEGG analyses are performed using ModEnrichr (https://maayanlab.cloud/modEnrichr/, accessed on 6 January 2023). Protein-protein interaction network analysis integrates Cytoscape 3.7.2 and the STRING database version 12.0 (http://string-db.org/), accessed on 10 April 2023.

In Vivo Experiments

Approximately 0.1 g of kidney tissue was cut and mixed with tissue lysis buffer (1:9 *w*/*v*). The mixture was lysed on ice for 10 min, homogenized in a mortar, and centrifuged at 10,000 × *g* at 4 °C. Synovial tissue was weighed, minced, and processed in a similar manner. HPRT1 protein in kidney tissue and CTSD protein in synovial and joint fluids were detected using the ELISA method.

In Vivo Experiments

NRK cells were thawed, cultured in 25 cm^2^ flasks (37 °C, 5% CO_2_), and passaged with 0.25% trypsin at 80–90% confluency. Cells were seeded in 6-well plates (1 × 10^5^ cells/mL), treated with the control/model group exosome-conditioned medium (2 mL/well, n = 3), and cultured for 24 h. Cells were lysed and stored at −80 °C. Protein quantification (BCA assay): Standards were diluted, mixed with working reagent (BCA Reagent: Cu2+ Reagent = 50:1), incubated (37 °C for 15–20 min), and measured at 570 nm. Western blotting was performed to detect CTSD and HPRT1 levels. 

#### 4.2.4. Statistical Analysis

Statistical analysis was performed using SPSS 20.0 software. Measurement data are expressed as mean ± standard deviation (*x* ± *s*). For comparisons among multiple groups, either one-way ANOVA or the Kruskal–Wallis H nonparametric test was selected based on whether the data followed a normal distribution. For pairwise comparisons between groups, either Dunnett’s t-test or Dunnett’s T3 test was used depending on the homogeneity of variance. Differences were considered statistically significant at *p* ≤ 0.05.

## 5. Conclusions

In summary, this study provides new insights into the pathological role of exosomes in gouty arthritis (GA) through comprehensive proteomic analysis and experimental validation. Using a GA rat model, we identified 40 significantly differentially expressed proteins (12 upregulated and 28 downregulated) and 56 group-specific proteins (16 upregulated and 40 downregulated). These proteins are enriched in key pathways, such as complement and coagulation cascades, autophagy, lysosomal function, and purine metabolism. Notably, tissue proteinase D (CTSD) expression was elevated in the model group, whereas hypoxanthine–guanosine phosphoribosyltransferase (HPRT1) expression was reduced, indicating that exosomes regulate key processes in the core pathological mechanisms of GA, including purine metabolism and lysosomal dysfunction. Further in vitro and in vivo experiments confirmed these findings; exosomes from GA model rats significantly modulated the expression of CTSD and HPRT1 in receptor cells. These results highlight the potential of exosomes as biomarkers and therapeutic targets for GA. Future research should focus on exploring the specific signaling pathways regulated by exosomes, such as those related to CTSD and HPRT1. Expanding the sample size and incorporating more inflammatory models would enhance the reliability and generalizability of the findings. This work not only deepens our understanding of the pathogenesis of GA, but also lays the foundation for innovative therapeutic strategies targeting exosome-mediated pathways.

## Figures and Tables

**Figure 1 ijms-26-05320-f001:**
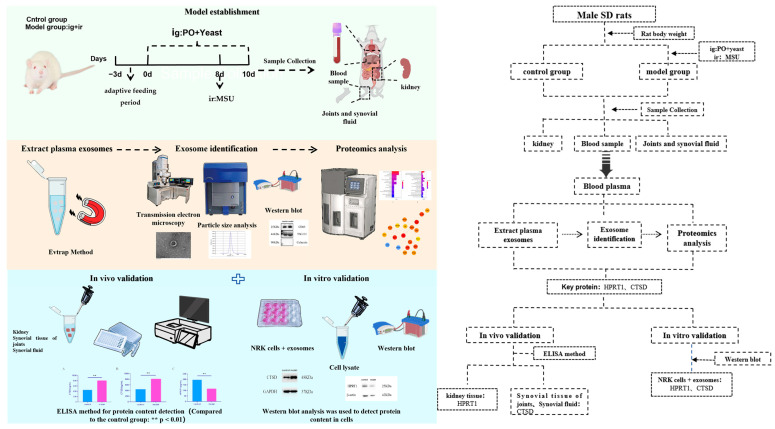
The flow chart of this experiment.

**Figure 2 ijms-26-05320-f002:**
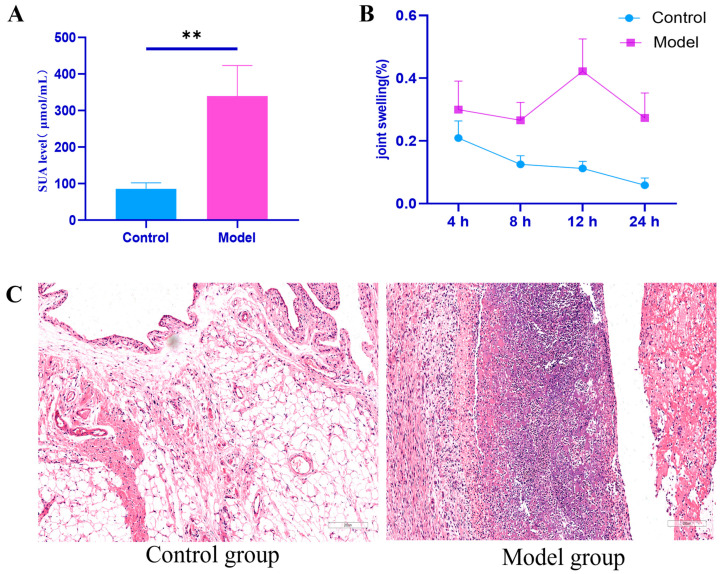
Establishment of rat model with hyperuricemia and gouty arthritis. (**A**) Serum uric acid levels in groups of rats (*n* = 12, ** *p* < 0.01); (**B**) changes in joint swelling in groups of rats; and (**C**) histopathological picture of synovial tissue in the ankle joint. Note: Blue represents the control group, purple represents the model group.

**Figure 3 ijms-26-05320-f003:**
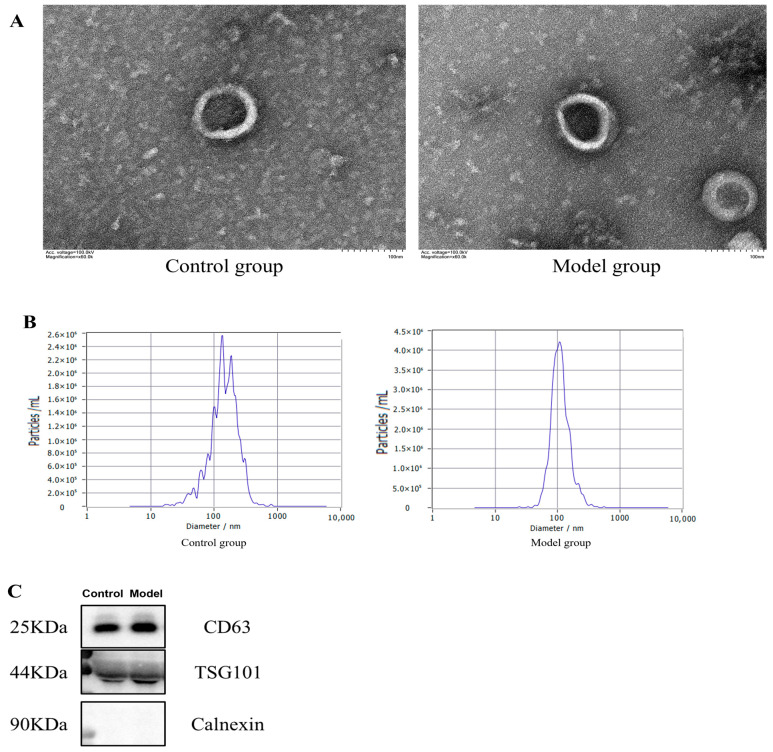
Characterization of plasma exosomes in rats. (**A**) Transmission electron microscopy results of plasma exosomes (scale bar: 100 nm); (**B**) schematic diagram of plasma exosome particle size concentration; and (**C**) immunoblot results for exosome marker proteins in each group.

**Figure 4 ijms-26-05320-f004:**
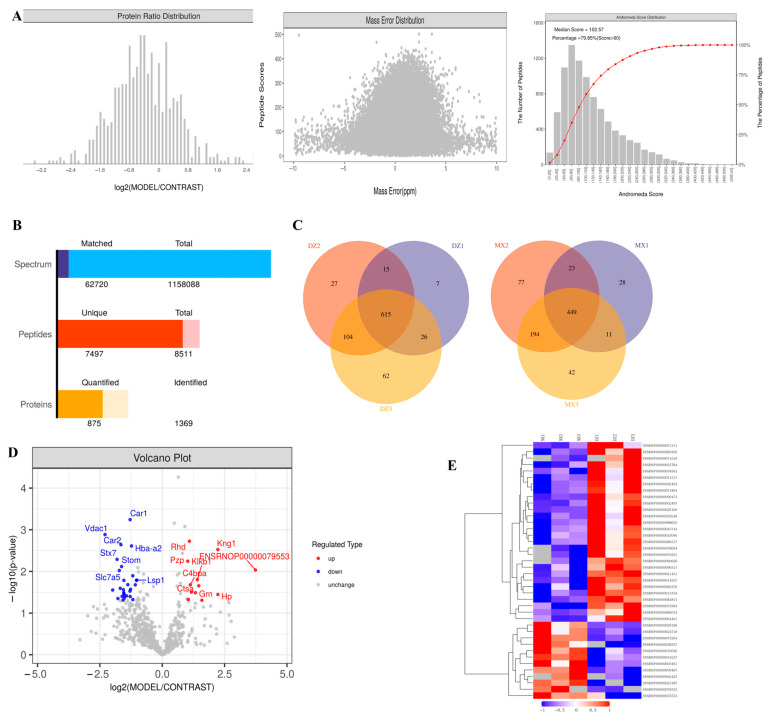
Proteomics analysis of plasma exosomes in rats. Note: The red line in the figure represents the cumulative curve, corresponding to the secondary vertical axis, indicating the cumulative percentage of proteins with a relative molecular mass not exceeding the corresponding value. (**A**) The quality control results of mass spectrometry analysis (protein abundance ratio distribution (CONTRAST—control group; MODEL—model group), peptide ion mass deviation distribution plot, and peptide ion score distribution plot). (**B**) Identification and quantitative results statistics. (**C**) Venn diagram of proteins identified by repeated identification in samples (DZ—control; MX—model). Note: Different colors represent three distinct samples within the group. (**D**) Protein quantification volcano map. (Note: The horizontal coordinate represents the fold change (log2-transformed), and the vertical coordinate represents the significance of the difference in *p*-values (log10-transformed). The red dots in the figure indicate significantly upregulated differentially expressed proteins, the blue dots indicate significantly downregulated differentially expressed proteins, and the gray dots represent proteins with no differential changes. (**E**) Cluster analysis of differentially expressed proteins.

**Figure 5 ijms-26-05320-f005:**
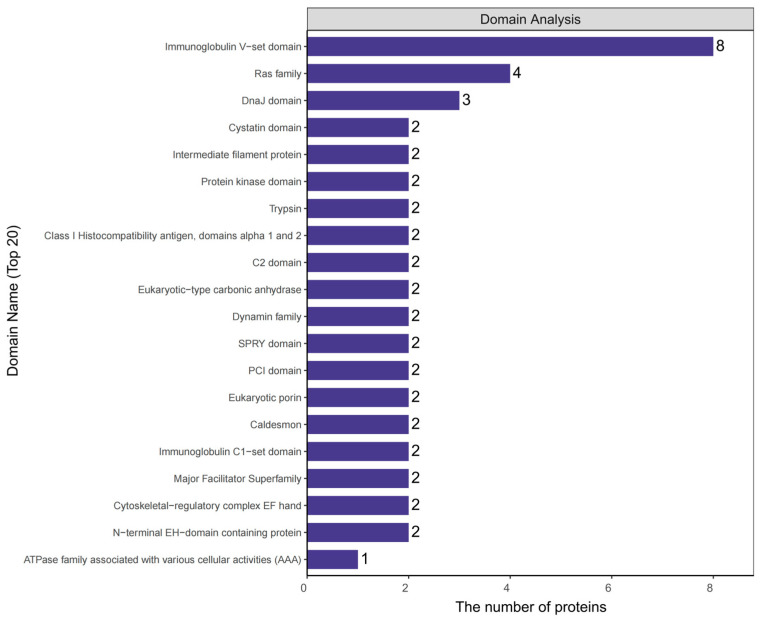
Analysis of structural domains of differentially expressed proteins. Note: The figure shows the domain (Domain Name) and its corresponding number of proteins (The number of proteins).

**Figure 6 ijms-26-05320-f006:**
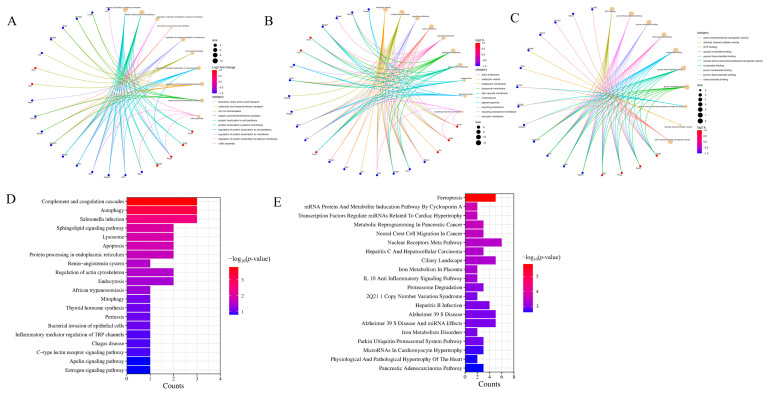
GO annotation and KEGG enrichment analysis of exosome proteins in rat plasma. (**A**) BP enrichment results; (**B**) CC enrichment results; (**C**) MF enrichment results; and (**D**) upregulated protein enrichment results and (**E**) downregulated protein enrichment results.

**Figure 7 ijms-26-05320-f007:**
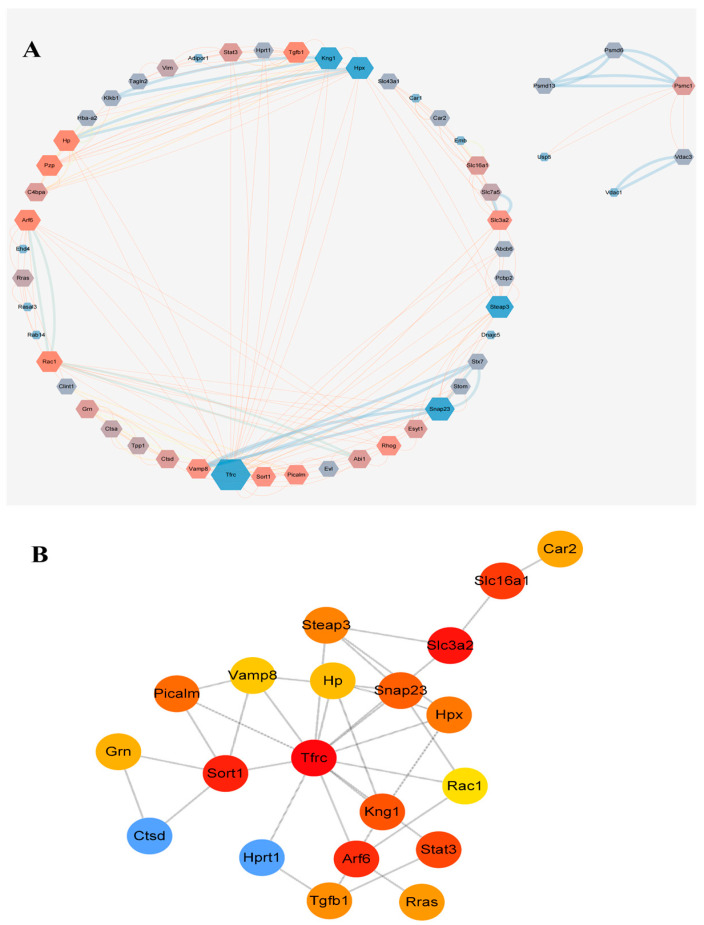
Protein interaction network analysis of exosome proteins in rat plasma. (**A**) Protein interaction network. (**B**) Interaction network of the top twenty core targets.

**Figure 8 ijms-26-05320-f008:**
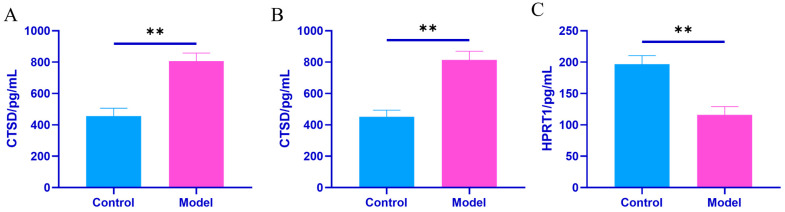
Protein expression in tissues and joint fluid. (**A**) CTSD level in synovial tissues in each group of rats, (**B**) CTSD level in joint fluid in each group of rats, and (**C**) HPRT1 expression in renal tissue of each group of rats. Note: Compared to the control group: ** *p* < 0.01.

**Figure 9 ijms-26-05320-f009:**
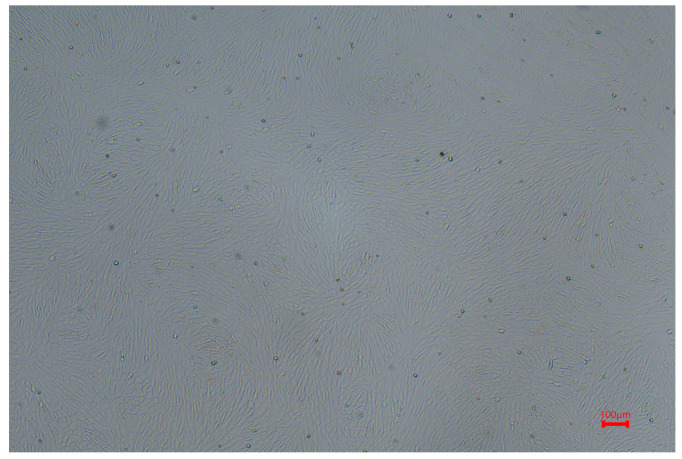
Morphology of NRK cells at 2 d~3 d (×40, scale bar 100 μm).

**Figure 10 ijms-26-05320-f010:**
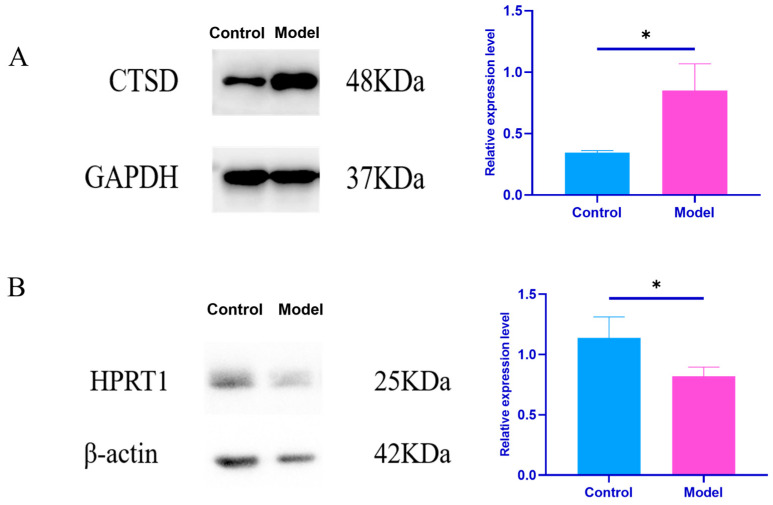
Relative protein expression in cells. (**A**) CTSD relative protein expression. (**B**) HPRT1 relative protein expression. Note: Compared to the control group: * *p* < 0.05.

**Table 1 ijms-26-05320-t001:** Changes in degree of joint swelling in each group of rats (*n* = 12, *x* ± *s*, unit: %).

Group	4 h	8 h	12 h	24 h
Control group	0.21 ± 0.05	0.13 ± 0.03	0.11 ± 0.02	0.06 ± 0.02
Model set	0.30 ± 0.09 **	0.27 ± 0.05 **	0.42 ± 0.10 **	0.27 ± 0.08 **

Note: Compared to the control group: ** *p* < 0.01.

**Table 2 ijms-26-05320-t002:** Exosome particle size detection results for each group (*n* = 3).

Group	Average Grain Diameter (nm)	Concentration (Particles/mL)
Control group	160.1	6.8 × 10^10^
Model set	118.6	2.2 × 10^10^

**Table 3 ijms-26-05320-t003:** Significantly differentially expressed plasma exosomal proteins.

Protein ID	Protein Name	Gene Name	*p* Value	Regulated Type
ENSRNOP00000079553			0.009234265	Up
ENSRNOP00000020196	Haptoglobin	Hp	0.035755848	Up
ENSRNOP00000075264	Kininogen 1	Kng1	0.002991341	Up
ENSRNOP00000024710	Hemopexin	Hpx	0.049060005	Up
ENSRNOP00000070092			0.02210595	Up
ENSRNOP00000019237	Kallikrein B1	Klkb1	0.015746642	Up
ENSRNOP00000028557	Granulin precursor	Grn	0.032365177	Up
ENSRNOP00000062425	Cathepsin A	Ctsa	0.03134781	Up
ENSRNOP00000005461	Complement component 4 binding	C4bpa	0.020942102	Up
ENSRNOP00000023187	Rh blood group, D antigen	Rhd	0.001881085	Up
ENSRNOP00000075573	Kininogen 2	Kng2l1	0.046996226	Up
ENSRNOP00000009467	Pregnancy zone protein	Pzp	0.005680288	Up
ENSRNOP00000096511	Lymphocyte-specific protein 1	Lsp1	0.016146578	Down
ENSRNOP00000090626	Voltage-dependent anion channel 3	Vdac3	0.016342616	Down
ENSRNOP00000078694	Ubiquitin-specific peptidase 5	Usp5	0.020973216	Down
ENSRNOP00000086034	Solute carrier family 43 member 1	Slc43a1	0.047359279	Down
ENSRNOP00000090473	Ubiquitin-40S ribosomal protein S27a-like		0.012683047	Down
ENSRNOP00000041451	Hemoglobin alpha, adult chain	Hba-a2	0.002447625	Down
ENSRNOP00000002407	Transferrin receptor	Tfrc	0.026856811	Down
ENSRNOP00000025649	RAB14, member RAS oncogene family	Rab14	0.040039527	Down
ENSRNOP00000014267	Carbonic anhydrase 1	Car1	0.000571536	Down
ENSRNOP00000071804	RT1 class Ia, locus A1	RT1-CE11	0.029687156	Down
ENSRNOP00000025196	Solute carrier family 3 member 2	Slc3a2	0.020725478	Down
ENSRNOP00000035090	Biliverdin reductase B	Blvrb	0.038578949	Down
ENSRNOP00000071616	Rac family small GTPase 1	Rac1	0.043284863	Down
ENSRNOP00000074317	Solute carrier family 44 member 1	Slc44a1	0.036478675	Down
ENSRNOP00000082579	Similar to Tpi1 protein	RGD1563601	0.016361424	Down
ENSRNOP00000078563	Embigin	Emb	0.033908058	Down
ENSRNOP00000064401	Synaptosome associate	AABR07053516.1	0.032261476	Down
ENSRNOP00000092404	Solute carrier family 16 member 1	Slc16a1	0.027531761	Down
ENSRNOP00000043743	Butyrophilin like 10	Btnl10	0.048835568	Down
ENSRNOP00000073171	ATP binding cassette subfamily	Abcb6	0.039659617	Down
ENSRNOP00000083562	Stomatin	Stom	0.007628371	Down
ENSRNOP00000013354	Carbonic anhydrase 2	Car2	0.002274499	Down
ENSRNOP00000086517	STEAP3 metalloreductase	Steap3	0.02542084	Down
ENSRNOP00000025784	Solute carrier family 7 member 5	Slc7a5	0.009520689	Down
ENSRNOP00000079201	Proline-rich coiled-coil 1	Prrc1	0.044639932	Down
ENSRNOP00000073994	Syntaxin 7	Stx7	0.005132134	Down
ENSRNOP00000088625	Vesicle-associated membrane protein 8	Vamp8	0.027923844	Down
ENSRNOP00000083811	Voltage-dependent anion channel 1	Vdac1	0.001304447	Down

**Table 4 ijms-26-05320-t004:** Protein expression in tissues and synovial fluid of rats in each group (*n* = 12, *x* ± *s*).

Group	Synovial Tissue CTSD (pg/mL)	Joint Fluid CTSD (pg/mL)	Kidney Tissue HPRT1 (pg/mL)
Control group	457.40 ± 46.51	451.11 ± 40.60	196.74 ± 13.10
Model set	806.50 ± 49.58 **	813.81 ± 53.19 **	115.98 ± 12.61 **

Note: Compared to the control group: ** *p* < 0.01.

**Table 5 ijms-26-05320-t005:** Protein expression of CTSD and HPRT1 in NRK cells (*n* = 3, *x* ± *s*).

Group	CTSD/GAPDH	HPRT1/β-Actin
Control group	0.34 ± 0.01	1.14 ± 0.15
Model set	0.85 ± 0.18 *	0.82 ± 0.07 *

Note: Compared with the control group: * *p* < 0.05.

**Table 6 ijms-26-05320-t006:** Exosome isolation procedure.

Order of Holes	1	2	3	4	5	6
Reagents added	Washing buffer containing magnetic beads	Washing buffer	1 mL PBS	1 mL PBS	200 μL 200 mM TEA solution elution	
	Drain the liquid 1	Remove liquid 2	Drain the liquid 3	Remove liquid 4	Merge eluates and store	

## Data Availability

Data will be made available on request.

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
