# Peer review of "Exploring Exosome Contributions to Gouty Arthritis: A Proteomics and Experimental Study"

_ijms, 2025, doi:10.3390/ijms26115320_

Round 1
Reviewer 1 Report
Comments and Suggestions for Authors
The manuscript presents a proteomics-based investigation into the role of plasma-derived exosomes in gouty arthritis, identifying CTSD and HPRT1 as potential mediators of disease via modulation of purine metabolism and apoptosis. While the study offers valuable initial insights.
- The authors used “Calnexin” as a control for purity, but the authors didn’t include so the author should include any cell lysate as the positive control.
- The authors claim that exosomes regulate purine metabolism and apoptosis in recipient cells via CTSD and HPRT1, but no mechanistic link is demonstrated.
- Functional assays assessing apoptosis, metabolic activity, and additional statistical rigour would strengthen the conclusions.
Author Response
The manuscript presents a proteomics-based investigation into the role of plasma-derived exosomes in gouty arthritis, identifying CTSD and HPRT1 as potential mediators of disease via modulation of purine metabolism and apoptosis. While the study offers valuable initial insights.
- The authors used “Calnexin” as a control for purity, but the authors didn’t include so the author should include any cell lysate as the positive control.
- The authors claim that exosomes regulate purine metabolism and apoptosis in recipient cells via CTSD and HPRT1, but no mechanistic link is demonstrated.
- Functional assays assessing apoptosis, metabolic activity, and additional statistical rigour would strengthen the conclusions.
Response to Reviewer 1:
Dear Reviewer:
We would like to express our sincere gratitude to the reviewers for their careful reading and valuable comments on our manuscript. We have carefully considered each of the reviewers' suggestions and have made the necessary revisions to improve the quality of our manuscript. Below is our point-by-point response to the reviewers' comments.
Respond to comment 1: Firstly, thank you for your question. Regarding the issue of "using Calnexin as a negative control to test the purity of exosomes but not using cell lysate as a positive control," our experimental design is based on the International Guidelines for Exosomes published by the International Society for Exosome Research (ISEV) and recommendations from relevant literature. When testing the purity of exosomes, we used the exosome marker proteins CD63 and TSG101 as positive controls.CD63: This is a transmembrane protein that is widely distributed in the endoplasmic reticulum, Golgi apparatus, and cell membrane. Inside cells, CD63 can be packaged into exosomes. Therefore, detecting CD63 is one of the important markers of exosome presence. TSG101: This is a key protein in the endoplasmic reticulum-Golgi intermediate (ESCRT) pathway. The ESCRT pathway plays a crucial role in the formation of intracellular vesicles, and the formation of exosomes also depends on this pathway. TSG101 participates in the biosynthesis process of exosomes, and its detection in exosomes indicates that the exosome formation process is normal. The formation of exosomes is closely related to the ESCRT pathway, and TSG101's involvement in the biosynthesis process of exosomes, as detected in isolated exosomes, suggests that the exosome formation process is normal. Additionally, we use Calnexin as a negative control. Calnexin is an endoplasmic reticulum resident protein that normally does not appear in exosomes. If Calnexin is detected during the assay, it may indicate contamination from cell debris or other impurities. Therefore, the presence or absence of Calnexin can serve as an important indicator for assessing the purity of exosome isolation. As for the cell lysate, it is typically used to extract all proteins within the cell, including those in the endoplasmic reticulum, cytoplasm, and cell membrane. Although the cell lysate can be used as a positive control, it contains too many types of proteins, which might mask the detection results of exosome-specific proteins. Therefore, using cell lysis buffer as a positive control may not accurately reflect the purity and characteristics of exosomes. Instead, using exosome marker proteins CD63 and TSG101 as positive controls and Calnexin as a negative control can more precisely assess the presence and purity of exosomes. These marker proteins are specific to the formation and secretion of exosomes and are widely recognized as indicators for exosome identification. In the revised manuscript submitted, we also added relevant references.
Respond to comment 2: Thank you for your question. Regarding the issue that "The authors claim that exosomes regulate purine metabolism and apoptosis in recipient cells via CTSD and HPRT1, but no mechanistic link is demonstrated," this study primarily focuses on establishing a rat model of hyperuricemia accompanied by acute gouty arthritis and extracting plasma exosomes from rats. The aim is to observe the differences in proteins carried by plasma exosomes between normal rats and model rats, followed by selecting two proteins from the core proteins that are closely associated with gout in existing studies for in vivo and in vitro experimental validation. Based on existing literature, it is suggested that CTSD and HPRT1 proteins can participate in purine metabolism and apoptosis, respectively. Thus, we preliminarily hypothesize that exosomes regulate purine metabolism and apoptosis in recipient cells through CTSD and HPRT1. In the revised manuscript submitted, I have reorganized the discussion and conclusions to more subtly indicate that exosomes regulate purine metabolism and apoptosis in recipient cells via CTSD and HPRT1.
Respond to comment 3: hank you for your question. Regarding the comment "Functional assays assessing apoptosis, metabolic activity, and additional statistical rigour would strengthen the conclusions," this study primarily focuses on investigating the protein differences in plasma exosomes between normal rats and gout rats. Based on existing literature, we selected CTSD protein and HPRT1 protein for preliminary validation. The experimental results consistently showed significant differences in the protein expression of CTSD and HPRT1 in gout rats compared to normal rats. Combined with existing research, we hypothesize that exosomes may influence the protein expression of CTSD and HPRT1 to regulate purine metabolism and apoptosis in recipient cells, thereby participating in the pathogenesis of gout. The main objective of this experiment is to conduct preliminary validation of proteomics findings in conjunction with existing research, proposing the pathological basis of exosome involvement in gouty arthritis. We sincerely appreciate your feedback, which has prompted deeper reflection on our experimental design.
We have made every effort to address the reviewers' comments comprehensively and believe that our manuscript has been significantly improved as a result. We have attached the revised manuscript along with this response for your convenience. We hope that the changes meet the reviewers' expectations and look forward to your favorable consideration for publication.
Reviewer 2 Report
Comments and Suggestions for Authors
The authors present an intriguing research manuscript.
However, authors are recommended to improve specific points, which are essential when writing and submitting a manuscript:
1. Authors are recommended to read the journal's guidelines to present their work properly. The title should also be restructured as it is too long and inappropriate.
2. Abstract: Remove the word "objective." It is not appropriate to add the words "Methods" and "Results." It is also recommended that the authors add a short introduction.
The introduction section is too limited. It is recommended to start with the general problem, then talk a little about the topic, then add and/or contrast with recent works of research related to the topic, and finally, or in the last paragraph, describe their contribution and objective of the research.
In addition, authors are advised not to use long sentences or paragraphs. Use good scientific writing.
4.- The second section is materials and methods, and the third section is results and discussion.
5. Figures 1 and 2 need to be labeled better and clearly. Their representations ( time /h) ( days/ W) are inappropriate. The model and control label should be inside the image or in an appropriate way, Not on the other side.
6. TEM images should be presented better and placed in clearer micrographs with appropriate scales.
7.- Images should be of better quality and presented correctly.
The results should be properly described, contrasted with research related to the work, and conclusive in each case.
The section on materials and methods should correctly describe all the steps. Likewise, it is recommended that a graphic abstract of the experimental setup be added.
9. The conclusion is very poor. The authors do not reflect on or highlight their research, which is not adequate or appropriate because it is very limited.
10.- Other questions
a) How can we be sure that the change in CTSD and HPRT1 is specific to exosomal signaling and not a general systemic effect of hyperuricemia?
b) Have you compared these proteomic profiles with exosomes isolated from a different model of joint inflammation (e.g., Freund's induced arthritis)?
c) Could they report effect size alongside P values to highlight the biological relevance of 12 protein changes?
d) What interindividual variability did they observe in exosome abundance (concentration and size) among the 12 animals, and how does this affect statistical power?
e) Although CTSD and HPRT1 change consistently in both models, is there any discrepancy in the magnitude of those changes that require explanation (e.g., 2× differences in vivo vs. 5× in vitro)?
Author Response
Reviewer 2
The authors present an intriguing research manuscript.
However, authors are recommended to improve specific points, which are essential when writing and submitting a manuscript:
- Authors are recommended to read the journal's guidelines to present their work properly. The title should also be restructured as it is too long and inappropriate.
2. Abstract: Remove the word "objective." It is not appropriate to add the words "Methods" and "Results." It is also recommended that the authors add a short introduction.
The introduction section is too limited. It is recommended to start with the general problem, then talk a little about the topic, then add and/or contrast with recent works of research related to the topic, and finally, or in the last paragraph, describe their contribution and objective of the research.
In addition, authors are advised not to use long sentences or paragraphs. Use good scientific writing.
4.- The second section is materials and methods, and the third section is results and discussion.
- Figures 1 and 2 need to be labeled better and clearly. Their representations ( time /h) ( days/ W) are inappropriate. The model and control label should be inside the image or in an appropriate way, Not on the other side.
- TEM images should be presented better and placed in clearer micrographs with appropriate scales.
7.- Images should be of better quality and presented correctly.
The results should be properly described, contrasted with research related to the work, and conclusive in each case.
The section on materials and methods should correctly describe all the steps. Likewise, it is recommended that a graphic abstract of the experimental setup be added.
- The conclusion is very poor. The authors do not reflect on or highlight their research, which is not adequate or appropriate because it is very limited.
10.- Other questions
- a) How can we be sure that the change in CTSD and HPRT1 is specific to exosomal signaling and not a general systemic effect of hyperuricemia?
- b) Have you compared these proteomic profiles with exosomes isolated from a different model of joint inflammation (e.g., Freund's induced arthritis)?
c) Could they report effect size alongside P values to highlight the biological relevance of 12 protein changes?
d) What interindividual variability did they observe in exosome abundance (concentration and size) among the 12 animals, and how does this affect statistical power? - e) Although CTSD and HPRT1 change consistently in both models, is there any discrepancy in the magnitude of those changes that require explanation (e.g., 2× differences in vivo vs. 5× in vitro)?
Dear Reviewer:
We would like to express our sincere gratitude to the reviewers for their careful reading and valuable comments on our manuscript. We have carefully considered each of the reviewers' suggestions and have made the necessary revisions to improve the quality of our manuscript. Below is our point-by-point response to the reviewers' comments.
Response 1.First of all, thank you for your advice. We have reviewed the journal's guidelines and restructured the title to make it more concise and appropriate. The new title reflects the core findings of our study more effectively.
Response 2. In the revised manuscript, we have carefully reorganized the abstract section according to your suggestions. Specifically, we have removed the word “objective” and eliminated the separate “methods” and “results” sections. Additionally, we have added a short introduction to provide essential background for the study.
Response 3. Regarding the introduction section, we have expanded it to start with a general problem statement, followed by a discussion of the topic. We have also incorporated recent research findings related to the topic, contrasting them where appropriate. Finally, we have clearly described our research contribution and objective in the last paragraph.Furthermore, we have reviewed the manuscript to ensure that we are using concise sentences and paragraphs, adhering to good scientific writing practices. We believe these changes have significantly improved the clarity and readability of our manuscript.
Response 4. Based on your proposal to organize the sections as "Materials and Methods" followed by "Results and Discussion," I revisited the journal's template and reviewed the framework of several articles recently published in the journal. Upon careful examination, I found that the journal actually requires a different structure: "Results" as the second section, "Discussion" as the third section, "Materials and Methods" as the fourth section, and "Conclusions" as the fifth section.Given this specific requirement, we have adhered to the journal's preferred structure to ensure consistency with their guidelines. We believe that following the journal's established format will facilitate the review process and align our manuscript with the expectations of the readership.Thank you once again for your input. We appreciate your attention to detail and will continue to carefully review our manuscript to meet the journal's standards.
Response 5.Thank you for your advice regarding the labeling of Figures 1 and 2. In response to your comments, we have thoroughly revised the labeling of these figures to enhance clarity and accuracy. Specifically, we have: Improved the Representation of Units: The labels for time and other units have been revised to ensure they are clear and appropriate. We have used standard notation and ensured that the units are correctly formatted and easily understandable. Repositioned Model and Control Labels: The labels for “model” and “control” have been moved from the side of the figures to within the images themselves. This change ensures that the labels are more prominently displayed and directly associated with the relevant data, improving the overall readability and interpretation of the figures.
Response 6.The transmission electron microscopy (TEM) images have been enhanced for clarity, and we have ensured that appropriate scales are included in the micrographs.
Response 7. We have reviewed and improved the quality of all images presented in the manuscript to ensure they meet the journal's standards.Results Description and Contrast with Related Research: We have revised the Results section to provide a more thorough description of our findings.
Response 8. We have comprehensively revised the Materials and Methods sections, providing as detailed a description of the experimental procedures as possible, and supplemented the graphical abstract in the submitted revised manuscript to enable readers to understand the research content more clearly and intuitively.
Response 9. The conclusion has been expanded to better reflect the implications of our research findings and to emphasize their relevance to the understanding of gouty arthritis.
Response 10. Additional Questions:
a) Regarding "How can we be sure that the change in CTSD and HPRT1 is specific to exosomal signaling and not a general systemic effect of hyperuricemia?" This study focuses on observing the differential expression of plasma exosomal proteins between normal rats and gout model rats. While CTSD and HPRT1 proteins may not be specific to exosomal signaling, the results show that these differentially expressed proteins in plasma exosomes of gout rats and normal rats include CTSD and HPRT1, indicating their important roles in exosome-mediated gout pathogenesis. Moreover, the correlation between CTSD, HPRT1 and hyperuricemia/gout has been partially reported in existing literature, as reflected in the revised manuscript submitted.
b). Regarding "Have you compared these proteomic profiles with exosomes isolated from a different model of joint inflammation (e.g., Freund's induced arthritis)?" We have not yet compared proteomics features with exosomes isolated from different joint inflammation models, as the focus of our study is on the role of exosomes in gouty arthritis, without considering other joint inflammation models. Additionally, we have reviewed relevant literature but have not found any related studies yet. Thank you again for your question,Your question has indeed prompted us to think more deeply about the broader implications of our findings. While our current study is limited to gouty arthritis, we recognize the value of comparing our results with those from other models of joint inflammation. This could provide valuable insights into the commonalities and distinct features of exosomal signaling in different inflammatory contexts.
- c) Thank you for your question. The relevant part has been added in the revised manuscript submitted.
- d) Regarding the inquiry about "what individual differences exist in exosome abundance (concentration and particle size) among the 12 animals, and how do these differences affect statistical power?", we used SD rats, which have relatively stable animal strains, and strictly standardized feeding conditions to minimize differences among individuals within the same group. Therefore, the exosome abundance among rats in the same group exhibits minor variations. However, there are certain differences in exosome distribution between different groups, which may be related to modeling conditions and other factors. Additionally, exosome abundance is closely associated with extraction conditions.
- e) Regarding the query "Although CTSD and HPRT1 change consistently in both models, is there any discrepancy in the magnitude of those changes that require explanation (e.g., 2× differences in vivo vs. 5× in vitro)?", no discrepancies in the magnitude of changes were observed in this experimental study.
We have made every effort to address the reviewers' comments comprehensively and believe that our manuscript has been significantly improved as a result. We have attached the revised manuscript along with this response for your convenience. We hope that the changes meet the reviewers' expectations and look forward to your favorable consideration for publication.
Thank you once again for giving us this opportunity to improve our manuscript.
Round 2
Reviewer 2 Report
Comments and Suggestions for Authors
The discussion is well grounded scientifically, employs modern techniques and presents a coherent narrative, l. However, to fully comply with the standards of the best journals, it is recommended to polish the style, improve the structural organization of the discussion, add critical depth, and better contextualize the findings clinically and with other research.
The discussion is well grounded scientifically, employs modern techniques and presents a coherent narrative, l. However, to fully comply with the standards of the best journals, it is recommended to polish the style, improve the structural organization of the discussion, add critical depth, and better contextualize the findings clinically and with other research.
- Revise style and grammar, shortening long sentences and eliminating repetitions or ambiguities.
Include more critical/comparative discussion of CTSD and HPRT1 in the pathophysiology of gout.
Correct minor formatting errors (such as “contributing to the of GA”).
Better contextualize findings within the clinical setting, e.g., implications for biomarkers or emerging therapies.
the title should be more specific and technical
Author Response
Reviewer
1.The discussion is well grounded scientifically, employs modern techniques and presents a coherent narrative, l. However, to fully comply with the standards of the best journals, it is recommended to polish the style, improve the structural organization of the discussion, add critical depth, and better contextualize the findings clinically and with other research.
The discussion is well grounded scientifically, employs modern techniques and presents a coherent narrative, l. However, to fully comply with the standards of the best journals, it is recommended to polish the style, improve the structural organization of the discussion, add critical depth, and better contextualize the findings clinically and with other research.
2.Revise style and grammar, shortening long sentences and eliminating repetitions or ambiguities.
3.Include more critical/comparative discussion of CTSD and HPRT1 in the pathophysiology of gout.
4.Correct minor formatting errors (such as “contributing to the of GA”).
Better contextualize findings within the clinical setting, e.g., implications for biomarkers or emerging therapies.
5.the title should be more specific and technical
Response to Reviewer:
Dear Reviewer:
We would like to express our sincere gratitude to the reviewers for their careful reading and valuable comments on our manuscript. We have carefully considered each of the reviewers' suggestions and have made the necessary revisions to improve the quality of our manuscript. Below is our point-by-point response to the reviewers' comments.
- We have thoroughly revised and reorganized the Discussion section. By incorporating the latest relevant literature, we have conducted a more in-depth analysis and reflection on our experimental results, adding critical depth and better contextualizing our findings within the clinical setting and in relation to another research.
- We have meticulously reviewed the entire manuscript, shortening long sentences, eliminating repetitions and ambiguities, and striving to clarify the language descriptions to ensure a more concise and precise presentation.
- Regarding the comment on including more critical/comparative discussion of CTSD and HPRT1 in the pathophysiology of gout: We have added a more detailed discussion on the physiological relevance of CTSD and HPRT1 proteins to gout in the Discussion section, integrating our experimental results with existing research to provide a comprehensive understanding.
- We have carefully reviewed the entire manuscript to correct formatting errors and minimize any potential issues. Additionally, we have revised the Discussion section to better align our experimental findings with clinical studies, highlighting their implications for biomarkers and emerging therapies.
- We have revised the article title to be more precise and reflective of the content.
In addition, I have thoroughly revised the Introduction, adding more background information and relevant references. The Results section has been reorganized and supplemented with related research findings, providing a more detailed description of our results. Corresponding revisions have also been made to the Conclusion and Abstract sections to ensure consistency and clarity.
We have made every effort to address the reviewers' comments comprehensively and believe that our manuscript has been significantly improved as a result. We have attached the revised manuscript along with this response for your convenience. We hope that the changes meet the reviewers' expectations and look forward to your favorable consideration for publication.
Thank you once again for giving us this opportunity to improve our manuscript.
Best regards,
Zhijian Lin